# Applications of Big Data and Blockchain Technology in Food Testing and Their Exploration on Educational Reform

**DOI:** 10.3390/foods13213391

**Published:** 2024-10-25

**Authors:** Haohan Ding, Zhenqi Xie, Chao Wang, Wei Yu, Xiaohui Cui, Zhenyu Wang

**Affiliations:** 1Science Center for Future Foods, Jiangnan University, Wuxi 214122, China; dinghaohan@jiangnan.edu.cn (H.D.); xcui@whu.edu.cn (X.C.); 2School of Artificial Intelligence and Computer Science, Jiangnan University, Wuxi 214122, China; 6223110049@jiangnan.edu.cn; 3Department of Chemical & Materials Engineering, University of Auckland, Auckland 1010, New Zealand; w.yu@auckland.ac.nz; 4School of Cyber Science and Engineering, Wuhan University, Wuhan 430072, China; 5Jiaxing Institute of Future Food, Jiaxing 314050, China; zhenyuwang@whu.edu.cn

**Keywords:** big data, blockchain, food testing, education reform

## Abstract

This study reviews the applications of big data (BD) and blockchain technology in modern food testing and explores their impact on educational reform. The first part highlights the critical role of BD in ensuring food safety across the supply chain, discussing various data collection methods, such as national and international food safety databases, while addressing the challenges related to data storage and real-time information retrieval. Additionally, blockchain technology has been explored for its ability to enhance transparency, traceability, and security in the food-testing process by creating immutable records of testing data, ensuring data integrity, and reducing the risk of tampering or fraud. The second part focuses on the influence of BD and blockchain on educational reform, particularly within food science curricula. BD enables data-driven curriculum design, supporting personalized learning and more effective educational outcomes, while blockchain ensures transparency in course management and credentials. This study advocates integrating these technologies into curriculum reform to enhance both the efficiency and quality of education.

## 1. Introduction

The World Health Organization defines food safety as the assurance that food is safe for its intended use and will not harm consumers when prepared and consumed [1]. To ensure this safety, food testing is performed frequently and routinely in the food industry for both quality assessment and safety assurance, leading to large datasets that may even be characterized as big data (BD). In recent years, food-related incidents have occurred frequently [2,3,4,5], with safety concerns being constantly reported worldwide. These reports have increasing credibility, international status, and impact on local online public opinion, underscoring the critical importance of food safety measures. These measures encompass the prevention, reduction, or elimination of physical, chemical, and biological hazard factors throughout the entire food chain—from production and processing to transportation, storage, sales, and consumption. Every step from the manufacturers to customers relies on food testing, and advancements in modern testing technology play a crucial role in mitigating the occurrence of food-related safety incidents [6].

Despite the critical role of food testing, there are several gaps in current research, particularly in the handling and analysis of the large volumes of data generated. Modern food testing not only requires precision, speed, and reliability, but also the ability to manage and analyze complex datasets from various stages of production. This paper aims to address these research gaps by exploring the role of BD in food testing processes, focusing on four key workflows: data collection, storage and transmission, analysis, and visualization. The study also delves into the challenges associated with these workflows, such as data silos, high data storage costs, and inefficiencies in data transfer and analysis.

This study also explored the application of blockchain technology in enhancing the management and security of data in food testing processes and its future prospects. While blockchain is not directly involved in conducting food tests, it plays a crucial role in supporting the infrastructure for data management within the food supply chain. Blockchain enhances transparency [7], security [8], and traceability by providing a decentralized and tamper-resistant ledger for real-time data recording. This ensures data integrity and secure sharing among stakeholders, such as producers, distributors, regulators, and retailers [9]. By using a decentralized and tamper-resistant ledger, the blockchain ensures the real-time recording of data, prevents tampering, and enables secure sharing among stakeholders, such as producers, distributors, regulators, and retailers [8]. It also supports scalable solutions for managing large volumes of testing data, while improving traceability and accountability in food safety management.

However, blockchains still face challenges in food testing. Key issues include ensuring secure data transmission, balancing information sharing among nodes, and improving the scalability of the blockchain to handle vast amounts of data. Techniques, such as sharding, are being explored to enhance the processing capacity [10]. Additionally, there is a need for improved compatibility and standardization across blockchain systems to ensure seamless data sharing and interaction within diverse food supply chains. Standardized protocols and formats are crucial to achieve this goal [11].

Furthermore, this study explores how the integration of BD and blockchain technologies is driving educational reform, particularly in food science and related curricula. These technologies have already demonstrated transformative applications across industries, including food testing, where blockchain ensures supply chain transparency and traceability, and BD enables efficient management and analysis of large-scale testing data. For instance, blockchain is used to monitor food safety from farm to fork, while BD supports predictive analytics in food safety management. As these technologies reshape operational practices and face numerous challenges in practical applications, educational institutions increasingly need to adjust their curricula to provide students with the skills and knowledge needed to manage and utilize these advanced tools. By integrating case studies and real-world applications into educational programs, students will be better prepared to address the complex challenges in modern food safety management and beyond. By bridging the gap between technological advancements and educational practices, we aim to contribute to the development of a workforce capable of leveraging these technologies effectively in the food industry.

### 1.1. Research Questions

This study aims to address the following research questions:How are big data and blockchain technologies currently being applied in food testing processes?What are the main challenges and future directions for implementing these technologies in food testing?How do the applications of big data and blockchain in food testing impact educational curricula in food science and related fields?

### 1.2. Research Design

This study employs a comprehensive literature review and case study analysis to explore the applications of big data and blockchain in food testing and their implications for educational reform. The research design consists of three main components:A systematic review of the academic literature and industry reports on big data and blockchain applications in food testing.An analysis of case studies demonstrating real-world implementations of these technologies in the food industry.An examination of the educational implications based on the identified technological trends and industry needs.

### 1.3. Hypotheses

Based on the initial literature review, we propose the following hypotheses:

**H1.** *The integration of Big Data and blockchain technologies in food testing processes significantly enhances food safety, traceability, and quality control*.

**H2.** *The implementation of these technologies in food testing faces significant challenges related to data management, security, and standardization*.

**H3.** *The widespread adoption of Big Data and blockchain in food testing necessitates substantial changes in food science and related educational curricula*.

### 1.4. Methodology

The methodology for this study involves the following:A literature review: A comprehensive review of peer-reviewed articles, industry reports, and relevant case studies published within the last ten years.A case study analysis: An examination of notable implementations of big data and blockchain in food testing across various segments of the food industry.A comparative analysis: Comparison of different technological approaches and their effectiveness in addressing food testing challenges.A gap analysis: Identification of gaps between current educational curricula and the skills required for implementing big data and blockchain in food testing.Synthesis: Integration of findings to develop recommendations for educational reform and future research directions.

## 2. Contributions, Challenges and Future Work in BD for Food Testing

De Mauro et al. [12] have proposed a consensus definition of BD: “Big Data represents the information assets characterized by a high Volume, Velocity, and Variety to require specific Technology and Analytical Methods for its transformation into Value”. With the application of intelligent devices, the characteristics of data fragmentation, distribution, and streaming media are more obvious. Mobile data are rapidly growing. The BD feature proposed by IBM adds veracity and value aspects. Figure 1 shows the stages in managing the workflow of a BD project and the tools used [13]. In each phase of a BD project, the contribution is not possible without the support of artificial intelligence (AI), such as edge computing, cloud computing, machine learning, and especially deep learning. In the next phase of this paper, we will explore the application of each phase in food testing, identify the challenges faced in each phase, and describe future work.

### 2.1. Data Collection: Online Food Testing Database (Early Warning System and Risk Assessment)

For online food testing databases, examples from different countries and organizations were collected in this study (Table 1).

A comprehensive overview of online food safety databases containing information on hazards, exposures, surveillance reports, and import refusal reports has been provided in the literature [14,15]. In the EU, the Rapid Alert System for Food and Feed (RASFF) remains the central online food safety database used by the authorities, industry, and scientists. In China, an official open-access food-testing database has not yet emerged. Only a few researchers have attempted to create an online food safety database using BD analysis techniques (data mining) [16,17,18]. Some commercial groups, such as food-mate (www.foodmate.net, accessed on 1 July 2024), update relevant food safety databases based on information published by the government.

### 2.2. Data Storage and Transferring: NoSQL Database and Social Media and Smartphone Case Studies

Unlike relational databases, such as MySQL (Version 8.3.0), Oracle (Version 23c), and PostgreSQL (Version 16.0), NoSQL (Version 7.2) is nonrelational. These SQL statements are open-source and horizontally scalable. The most typical example of NoSQL technology is the MongoDB database. The ability to efficiently and securely transfer BD is unrealistic, with only physical transfers, such as disks and CDs, and dedicated software, such as Aspera (Version 4.2.2) and Talend (Version 8.0), available to transfer and share BD.

In recent years, the growth of the Internet has resulted in its shift from being a potential source of food safety data [19] to a major way of obtaining and sharing food safety information in the public eye [20]. By analyzing social BD, social media has become an effective way to raise public awareness and food testing standards, and to update legislative standards, while strengthening the food market surveillance mechanism based on the trend of continuous publicization of social media data [21,22]. However, social media as a public platform has also become a tool for the dissemination of rumors and false information, particularly during the 2019 COVID-19 pandemic. These include rumors related to rapid antigen testing [23], misinformation about pharmaceuticals [24], and false claims regarding vaccine side effects [25]. Furthermore, the rapid dissemination and widespread nature of information on social media platforms may lead to personal opinions being misconstrued as professional food-testing reports or the instantaneous spread of food-testing information lacking a scientific basis. Users are often influenced by sensationalism and emotionally charged portrayals [26], predisposing them to form subjective opinions without comprehensive understanding [27]. Research indicates that large language models can significantly enhance false-text detection rates through baseline statistical analysis. As we anticipate a manifold increase in the quantity and diversity of news sources in the social media era, a novel distributed architecture incorporating blockchain technology for content verification has emerged [28].

Smartphones with integrated sensors, such as GPS, accelerometers, gyroscopes, cameras, and microphones, can be used for various purposes, such as quality assessment, food testing, monitoring, behavior management, and food safety information exchange [29]. Portable intelligent devices are often used as indirect testing tools or media for data transmissions. They can only be directly and automatically used for food inspection when inspecting small volumes of food. Transferring BD information from different smartphones or data sources to be processed in NoSQL clusters provides a wider and more trustworthy data source for the next step in data analysis. Alfian et al. [30] developed a real-time perishable food quality monitoring system that receives temperature, humidity, GPS, and image data from smartphones and stores it in the MongoDB database. When tested on the kimchi supply chain in South Korea, as the number of sensors and clients increases, the system can effectively process a large amount of sensor data input/output, thereby improving the efficiency of food monitoring.

### 2.3. Data Analysis: The Heart of BD Workflows

Data analysis plays a crucial role in the food testing industry, as it allows for extracting valuable insights from the large datasets generated during testing processes. After data is stored in SQL and moved to the processing unit, data analysis is performed, which is central to the value creation process, often referred to as the fifth “V” of BD—value [14]. Since 2019, various approaches (Table 2) have been used to extract knowledge from BD on food safety for food testing, including image processing algorithms for quality detection, data mining algorithms for consumer behavior analysis, statistical analysis for food safety monitoring, natural language processing for label compliance checking, time-series analysis for supply chain management, and deep learning techniques for allergen detection and harmful substance assessment.

### 2.4. Data Visualization

Data visualization tools can be used for forecasting demand, decision analysis, and so on. Data visualization tools do not directly demonstrate their usefulness for food testing. However, data visualization is essential as part of the BD process, where researchers can analyze food safety ingredient data to develop new or update existing food ingredient specifications, such as additive standards.

The internationally used data visualization tools were R (http://cran.r-project.org/, accessed on 5 July 2024), Tableau (http://www.tableausoftware.com/, accessed on 5 July 2024), and Circos (http://circos.ca/, accessed on 5 July 2024). Among these, R and Circos require users to have a certain programming foundation, which is challenging to start. Tableau is commercial software, and users do not need to have a programming foundation. The mainstream data visualization tools in China are Superset (https://superset.apache.org, accessed on 6 July 2024), DataEase (https://github.com/dataease/dataease, accessed on 6 July 2024), and Meta-Base (https://www.metabase.com, accessed on 6 July 2024). Again, Superset requires basic knowledge of the SQL language, while DataEase and Meta-base do not.

### 2.5. Challenges and Solutions Faced by the Four Stages of Food Testing BD Workflow

In this subsection, we explore the challenges and solutions corresponding to the four phases of BD mentioned above (from 2.5.1: Data Collection to 2.5.2: Data Storage and Transmission to 2.5.3: Data Analytics and finally to 2.5.4: Data Visualization).

#### 2.5.1. Low Data Collection Efficiency and Poor Data Quality

In certain special applications of food testing, such as the collection of testing BD for food microbiological testing, food chemical residue and contaminant testing, or food sensory evaluation, the collection process often relies on physical sampling or manual collection of data, resulting in limited access to advanced data collection technologies, such as IoT devices, sensors [45], and automatic data recording systems [46,47] which can both reduce transcription errors and allow a higher data frequency. Therefore, compared to fields that widely adopt advanced data collection techniques, the collection of food testing BD is still slow and inefficient.

Meanwhile, the diversity of BD sources in food testing not only leads to gaps and slowness in the data collection process but can also adversely affect the quality of data. Data quality is defined as the measurement of whether the data are suitable for its intended use. Food testing data can come from various sources, including links in the production chain [48], consumer feedback, online social media [49], and government reports. The sheer diversity of these sources means that the format and quality of the data are typically variable, leading to applicability and consistency issues when attempting to integrate such data. Furthermore, poor data quality leads to insufficient, inconsistent, and even erroneous decision-making by researchers, and existing research focuses on the development of data quality assessment frameworks. Fadlallah et al. [50] reviewed the existing research on situational awareness-based BD quality assessment and summarized an advanced methodological framework for situational awareness of BD quality assessment solutions. Elouataoui et al. [51] evaluated the quality of BD using weighted indicators based on 12 V characteristics extended from BD and different scenarios.

#### 2.5.2. Data Silos, High Data Storage Costs, and Inefficient Data Transfer

In the food testing sector, the independent management (non-sharing of data) of commercially sensitive information or data from private companies and important data from food regulatory authorities (which may involve state secrets) leads to a pile-up of food safety data and inhibits the flow of BD [52]. Privacy protection technology (PPT) refers to a collection of technologies and methods aimed at protecting the privacy of personal data, which, when combined with machine learning (ML, a branch of AI focused on enabling computers to learn from data and make decisions or predictions), can support BD sharing without compromising privacy. Integrating PPT with ML for BD sharing primarily involves training models with differential privacy, as described in [53], which masks individual data points to prevent identification while allowing the overall data trends to be analyzed. Data analysis using homomorphic encryption, as explained in [54], enables computations of encrypted data without requiring decryption, thereby preserving data confidentiality. The method of joint learning with secure multiparty computation, highlighted in [55], allows multiple parties to collaboratively analyze data without sharing the actual data itself, ensuring the privacy of each party’s data. Additionally, as an ML model that, like blockchain, adopts a decentralized architecture, federated learning possesses privacy protection advantages, such as high network bandwidth efficiency and adaptability to different data distributions [56]. This makes them particularly suitable for use in sensitive BD environments.

The volume, velocity, and variety of the 5 V characteristics of BD covered above pose storage challenges for distributed storage systems and blockchain-based cloud storage services [57]. Distributed storage systems and blockchain-based cloud storage services have a strong competitive market, but the following factors must be considered when storing BD: data consistency, data availability, and partition fault tolerance [58].

The life cycle of data transfer is divided into four stages, as shown in Figure 2, which was adapted from [59]. In each phase, there are significant challenges in transferring large amounts of data, as noted in [60]. For example, key challenges include ensuring data quality and timeliness during collection, handling data heterogeneity and scale during integration, addressing data security and transmission efficiency in management, and meeting preprocessing and resource optimization needs in the analysis phase. Identifying and addressing these challenges is crucial for the effective processing and analysis of BD. Therefore, data preprocessing is important. Yang et al. [61] used a MapReduce framework and data compression techniques to effectively reduce the size of unit data before transferring the data, and proposed an efficient network data transfer model that supports LZMA (Lempel–Ziv–Markov chain algorithm) and DEFLATE (combines Huffman coding and LZ77 coding) data compression/decompression techniques. The MapReduce framework is a programming model used to process large datasets across distributed clusters. It simplifies data processing on large scales by dividing tasks into smaller subtasks, The term ‘Map’ means that one processes and transforms individual data elements into key-value pairs, and the term ‘Reduce’ means to aggregate these pairs based on their keys to produce a consolidated output. LZMA (Lempel–Ziv–Markov chain algorithm) is a standard data compression algorithm known for high compression ratios, using a Lempel–Ziv method variant with Markov chain principles. DEFLATE combines LZ77, identifying repeated strings, with Huffman coding, which assigns shorter codes to more frequent data. Unlike lossy compression algorithms, lossless compression algorithms compress data without causing information loss.

#### 2.5.3. High Data Complexity and Challenge of Diversity in Data Analysis Methods

Understanding the relationship between features and developing effective data-mining methods that can accurately predict future observation results is an important goal in the data analysis phase [62]. However, as the volume of data from food-testing equipment continues to increase, analyzing BD poses challenges to the time complexity, spatial complexity, and scalability of the underlying algorithms [63]. For data analysis, Begoli and Hoery [64] first proposed three system design principles: First, the architecture should support multiple analytic methods, such as statistical analysis, ML, data mining, and visualization analysis. Second, a single storage mechanism should not be used, as there is no single style of database that can meet the needs of all types of data, and specialized data management systems are required for data of different sizes. In addition, the data should be handled differently at different stages. Third, it is important to make the data accessible, for example, by using open popular standards, lightweight architectures, and exposing results using APIs. To produce timely BD analytics results, computationally intensive data-mining algorithms and high-performance processors are required. Cloud computing infrastructure can serve as an effective platform to address the computational and data storage needs of BD analytics applications, and the trend of adoption is increasing [65].

#### 2.5.4. Difficulty in Real-Time Visualization of Unstructured Data

The goal of data visualization is to show data relationships more intuitively and effectively using different graphic technologies. In the process of transforming abstract data into intuitive graphics, visualization tools face the challenge of data processing, particularly in the production environment of rapid food testing. Food testing data are usually unstructured or semi-structured, and it is difficult to generate diverse graphics in real-time. In the future, with the exponential increase in data volume, the privacy problem of data and information visualization deserves attention. Andrienko et al. [66] proposed the idea of “federation visualization,” aiming to apply the federated learning algorithm in the field of AI to the field of data visualization. Simultaneously, the value (or availability) of visualized data continues to improve, and building a visualized graphical database that can process heterogeneous data in real time can help users map similar graphical styles to more efficiently process data relationships [67].

## 3. Future Work: Blockchain

Considering that the projected global population will surpass 9 billion by 2050, food security and a sustainable food supply chain will be required to meet global food demand [68]. Food testing should receive the attention of governments to ensure food safety from producers to customers. Food testing poses numerous challenges for BD technology, demanding effective strategies to handle the complexities of large-scale data. Successful processing and analysis of data within defined time frames necessitate a substantial computing infrastructure. Although cloud services, encompassing cloud computing and storage, have been widely embraced as solutions by various organizations, the exploration of cloud computing technologies for BD applications in food testing remains in its nascent stage. Several research challenges, including security, privacy, and legal concerns, are yet to be comprehensively addressed in this context.

Blockchain technology has great application prospects in food testing and is expected to produce a safer and more transparent food chain. Blockchain helps food testing service providers share data with other stakeholders, while reducing the risk of data leakage. In addition, storing food testing BD from different stages in the blockchain can eliminate duplicate food data analyses. Each test result is recorded on the blockchain, and the entire supply chain process, from the farm to the dining table, can be traced to ensure food safety and quality.

### 3.1. Blockchain Technology

A blockchain is a distributed, decentralized ledger technology developed through leading-edge technologies, including P2P networks, distributed ledgers, smart contracts, consensus, and cryptography, to ensure that transactions between multiple computers are recorded in a secure and transparent manner. Each “block” in the chain contains a number of transactions, and each time a new transaction occurs on the blockchain, a record of that transaction is added to each participant’s ledger. A block consists of the block version of the previous block, cryptographic hash, timestamp, nBits, nonce, transaction counter, and transaction data (Merkle tree) [69]. The decentralized nature of blockchain makes it resistant to data tampering because, once recorded, data cannot be retroactively altered unless all subsequent blocks are modified. This decentralized characteristic enhances trust and security in data transactions.

Originally developed for Bitcoin, blockchain technology has since been applied across various sectors. In supply chain management, it enhances product traceability and mitigates fraud [70]. In financial services, it facilitates secure cross-border transactions and supports decentralized finance [71]. In healthcare, blockchain is used for the secure management of electronic health records and pharmaceutical traceability [72]. The technology also plays a critical role in intellectual property protection, safeguarding digital rights and patents [73]. Within public services, blockchain enables secure electronic voting and digital identity verification [74]. In energy management, it supports peer-to-peer energy trading and carbon credit tracking [75], while in the charity sector, it increases transparency through donation tracking [76]. Furthermore, in the legal sector, blockchain-powered smart contracts reduce the need for intermediaries, ensuring efficient contract execution [77]. Among these applications, blockchain is also employed in the food-testing industry, where it monitors the supply chain from farm to fork, ensuring food safety and quality.

Blockchain can be classified into four different deployment modes based on various application scenarios: public [78], private [79], hybrid [80], and consortium [81]. In the field of food testing, private and consortium blockchains are more widely applied. A private blockchain is suitable for data management within a single entity or organization, aiding in tracking production and supply chain processes to ensure food safety. On the other hand, a consortium blockchain facilitates reliable sharing of testing data and supply chain information through joint control by multiple parties, such as farms, producers, distributors, retailers, and government regulatory agencies, thereby enhancing overall efficiency in food safety and quality management [82]. Currently, the application of blockchain technology in the food testing field primarily focuses on strengthening supply chain traceability, as shown in Table 3.

### 3.2. Open Issues and Future Directions

Owing to the complex scale of blockchain networks and their extensive resource consumption, coupled with the fact that they are not yet mature, there is still an urgent need for further scrutiny and emphasis on certain unresolved issues. The following are some open issues and future research directions.

Information security and accessibility: Against the backdrop of the global food chain, there is a growing need to focus on the security and trustworthiness of testing information. Therefore, the system requires standardized security protocols to address transaction and accessibility-related issues, providing security for consensus algorithms. Hyperledger Fabric is a permissioned blockchain platform, meaning that only authorized users can join the network, ensuring a controlled and secure environment [83]. Information security protocols are uniformly managed through chaincode (https://hyperledger-fabric.readthedocs.io/en/release-2.5/smartcontract/smartcontract.html, accessed on 17 October 2024), providing standardized mechanisms for managing transaction security and accessibility. In Hyperledger Fabric, channels play a key role in enhancing privacy and confidentiality. Channels are private “subnets” of communication between two or more specific network members, for the purpose of conducting private and confidential transactions [84]. Hyperledger Fabric allows developers to customize the number of channels based on design requirements, including single-channel [85], dual-channel [86], triple-channel [87], and multi-channel designs [88]. This feature allows for the segregation of data, ensuring that sensitive food testing information is only accessible to authorized parties. Future work should focus on optimizing the design and implementation of channels in blockchain systems for food testing. This includes exploring how to balance the need for data privacy with the requirement for transparency in food supply chains [89], and investigating how channel configurations can be optimized to enhance system performance and scalability while maintaining high levels of security [90].Information sharing: The proof-of-stake protocols in blockchain consistently result in nodes with more information gaining additional data and being selected for mining, leading to an imbalance in information sharing among the participating nodes in the blockchain network [10]. To address this issue, at the system level, a concept framework of fairness was proposed [91], providing all nodes in the blockchain network with equal opportunities and enabling every participant in the food chain to access fair testing information. At the technical level, Tao et al. [92], based on a cloud-fused BD blockchain, introduced a blockchain-cloud fusion solution based on decentralized attribute-based signatures (DABS), aiming to strengthen information sharing among different departments.Scalability of blockchain–data integration: The scalability performance of blockchain is measured based on transaction and data read throughput/latency, as well as data storage capacity. Blockchain can be scaled to any number of users without compromising data integrity and privacy. However, a prominent challenge with BD lies in the complexity of its nature, making the scalability of integrating blockchain with BD a significant and primary challenge for blockchain technology itself [93]. Sharding techniques aim to enhance overall network scalability by dividing the blockchain network into multiple independent fragments. Each shard is responsible for processing a portion of the data, operating independently of other shards, achieving horizontal scalability, and improving the throughput and performance. Future research can focus on optimizing sharding techniques and addressing the two common challenges faced by sharding. Inefficient sharding allocation schemes can lead to new issues related to data security and system scalability. Dhulavvagol et al. [94] adopted a hybrid sharding strategy to create multiple shards or partitions, thereby enhancing the scalability of the blockchain system. Xu et al. [95] designed a sharding scheme based on graph partitioning, which significantly balanced the shared distribution and reduced the data throughput latency. Another challenge is to protect data security within shards. Cai et al. [96] proposed a multi-objective objective optimization algorithm to enhance the security of large-scale testing data in the food supply chain. From a broader perspective, Li et al. [97] designed a blockchain combined with a PPT scheme applied at a higher level in a blockchain network.Compatibility and standardization: Due to the diversity of the food chain, various types of blockchains should be customized according to the specific characteristics of each food chain. Standardized blockchain protocols and data formats can provide a consistent interaction framework for different types of blockchains, facilitating data sharing and collaboration among regulatory authorities [11,98]. Additionally, the application of standardization helps enhance the system‘s credibility, prevent fraud, reduce barriers to adopting blockchain technology, and provide a more consistent workflow for all participants in the food chain. For different categories of a Food A testing BD, such as microscopic data on microbial indicators for Food A and macroscopic traceability information for Food A, microscopic microbial indicator data for Food A, and microbial indicator data for Food B, blockchains must address diverse group demands and be compatible with complex food testing BD. The design of an adaptive blockchain for BD is essential, with popular adaptive blockchain designs, including lightweight blockchains suitable for consumer real-time BD needs and scalable blockchains suitable for regulatory authorities handling large-scale testing data. Bandara et al. [99] summarized scalable adaptive blockchains using different consensus algorithms applied to various scenarios.
foods-13-03391-t003_Table 3Table 3Application cases of blockchain technology in food testing (food supply chain traceability).ReferencesResearch SubjectsExperimental ResultsArena et al. [100]Extra virgin olive oil supply chainProposed a blockchain-based Bruschetta traceability system that records data using a proposed system based on the Hyperledger Fabric platformLiu et al. [101]Imported fresh food supply chainTracked and detected fresh food information from source to destinationLu et al. [102]Food anti-counterfeit traceabilityProposed a blockchain and IoT-based food anti-counterfeiting traceability system, which uses AES encryption technology to encrypt it, and the system has higher security, lower transaction latency, and lower communication costBurgess et al. [103]Short food supply chainDeveloped a blockchain-based quality testing management architecture for the short food supply chainCao et al. [104]Australian beefMultiple signature approach based on STN and PoA blockchain for improved governance of geographically dispersed beef supply chainsBumblauskas et al. [105]Egg supply chainTracked and inspected eggs in the supply chain from farm to consumer, increasing efficiency by reducing the risk of food recalls, fraud and product damageDey et al. [106]Milk, pumpkinDigitized food production information in QR codes and made it easy for customers and producers to detect and verify, FoodSQRBlock was built using the Google Cloud PlatformCocco et al. [107]Italian Carasau breadProposed a smart contract-based blockchain that provided transparency and traceability for the Calabrian supply chain in ItalySalah et al. [108]Soybean supply chainUtilized the decentralized file system (IPFS) for a blockchain-based traceability system for the soybean supply chain on an Ethereum platform and smart contracts, to standardize the in-chain testing process and transaction managementKumar et al. [109]Rice supply chainAdopted a blockchain system for comprehensive traceability of the rice supply chain to combat food fraud, and implemented automation using smart contractsXie et al. [110]AppleProposed an integrated machine-to-machine traceability data generation system as an implementation of blockchain, to automatically access apple production information and enhance testing efficiencyYang et al. [111]FruitsA dual storage structure of "database + blockchain" was established, and a reputation-based smart contract was designed to ensure the authenticity and reliability of data in fruit and vegetable supply chain managementWang et al. [112]Fish supply chainA fish source and quality testing and tracking (BeFAQT) system was developed, enabled by blockchain, and a multi-layer blockchain architecture based on attribute encryption (ABE) was proposed to address the privacy issue caused by the application of blockchain in protecting supply chain data and achieving trusted and confidential data sharing among all parties in the fish supply chain


## 4. The Application of BD and Blockchain Technology in the Food Industry and Its Impact on Educational Curriculum Reform

BD and blockchain technology have become increasingly important in the food industry, improving efficiency in food safety and quality control and driving new business models and marketing strategies. These technologies also have profound implications for the field of education, particularly for the reform of food testing-related curricula. This section analyzes the specific applications of BD and blockchain technology in the food industry (as shown in Figure 3) and discusses how these applications inform and inspire educational curriculum reform.

### 4.1. Real-Time Food Safety Monitoring and Predictive Analytics

The adoption of BD analytics for real-time monitoring of food safety parameters throughout the supply chain presents both significant opportunities and challenges. While this approach allows for early detection of potential hazards and predictive analysis of food safety risks, it also demands a transformation in how food safety professionals are trained and educated.

Nestlé implemented a BD-driven quality assurance system that collects and analyzes data from over 100,000 suppliers. This system allows for real-time monitoring of food safety parameters and early detection of potential issues (https://www.nestle.com/csv/raw-materials, accessed on 25 September 2024). Recent research [113] has highlighted the importance of the Internet of Things (IoT) in food safety monitoring. Bouzembrak et al. [114] conducted a comprehensive literature review and bibliometric analysis, demonstrating the growing trend of IoT applications in food safety. Furthermore, Mishra et al. [115] proposed an IoT-based framework for food safety and quality monitoring in food processing industries, showcasing the potential of these technologies in real-world applications.

However, Ghashim et al. [116] pointed out that concerns about data security and privacy are one of the biggest obstacles to IoT adoption in the education sector, while also highlighting challenges in data standardization and system interoperability, highlighting the necessity of establishing industry wide protocols. Curriculum reform needs to focus on cultivating students’ abilities in data quality assessment, standardization methods, and heterogeneous data processing. The educational content should cover advanced data collection technologies, such as the application of IoT devices, sensors, and automatic data recording systems, to improve data collection efficiency and accuracy. In addition, given the importance of data privacy and security, the course needs to include the principles and applications of privacy protection technologies, such as differential privacy, homomorphic encryption, and federated learning. To tackle the challenges of large-scale data processing, students need to learn the use of distributed computing frameworks, such as MapReduce and cloud computing infrastructure. Finally, based on the demand for multi method data analysis, education should cultivate students’ ability to master various analysis methods, such as statistical analysis, machine learning, data mining, and visualization analysis.

### 4.2. Rapid and Non-Destructive Food Testing Techniques

BD and AI have facilitated the development of rapid, non-destructive food testing methods, such as hyperspectral imaging and electronic noses.

Advancements in portable spectroscopy, such as those developed by Ocean Optics, have enabled rapid, non-destructive analysis of food composition and quality. These devices generate large datasets that are analyzed using machine learning algorithms to determine food authenticity and quality [117]. The application of near infrared (NIR) spectroscopy in food production has been extensively studied. Porep et al. [118] reviewed the online application of NIR spectroscopy in food production, highlighting its potential for rapid and non-destructive analysis. Grassi and Alamprese [119] further discussed the advances in NIR spectroscopy applied to process analytical technology in food industries, emphasizing its role in ensuring food quality and safety.

Curriculum reform needs to focus on the integration of interdisciplinary knowledge, enabling students to understand and apply the principles of advanced detection technologies, such as hyperspectral imaging and electronic noses. Considering the unstructured or semi-structured nature of food testing data, students need to master advanced data visualization skills, especially real-time visualization methods for complex data. In addition, given the potential application of blockchain technology in food traceability, the course should include basic knowledge of blockchain technology, especially the use of licensed blockchain platforms, such as Hyperledger Fabric, as well as how to design and implement blockchain systems for food testing. To tackle the challenges of integrating big data and blockchain, students need to learn how to design and implement blockchain systems capable of handling large-scale data, including the application of sharding technology. Finally, considering the diversity of the food industry, the course should cover data standards and protocols in the food industry, as well as how to design adaptive blockchain systems that are compatible with different needs [120].

### 4.3. Blockchain for Food Traceability and Authenticity Verification

Blockchain technology provides an immutable record of a food product’s journey through the supply chain, enhancing traceability and authenticity verification. Walmart collaborated with IBM to implement the Food Trust blockchain platform, which allows for end-to-end traceability of food products. In one test, Walmart was able to trace the origin of a package of mangoes in 2.2 s, a process that previously took nearly 7 days [121]. Feng et al. [9] reviewed the development methods, benefits, and challenges of applying blockchain technology to improve agri-food traceability. Behnke and Janssen [11] investigated the boundary conditions for traceability in food supply chains using blockchain technology, providing insights into its practical implementation.

However, the energy consumption of blockchain systems [122], the lack of industry standards [123], and the complexity of integrating with existing supply chain technologies [93] are all issues that need to be addressed. Curriculum reform requires the integration of blockchain technology modules into traditional supply chain management courses, with a focus on its application in food traceability. Students should learn how to design and implement blockchain based traceability systems, including the development of smart contracts and the selection of consensus mechanisms. Meanwhile, the course should also cover relevant topics, such as data privacy and security, system integration, energy efficiency, etc., enabling students to fully understand the challenges and limitations of blockchain technology in the food industry. In addition, considering the global nature of the food supply chain, the curriculum should include content on international standards and regulations, as well as how to design blockchain systems that meet the needs of different regions.

### 4.4. IoT and Sensor Networks in Food Quality Monitoring

The Internet of Things (IoT) and sensor networks generate vast amounts of data on food quality parameters throughout production and distribution. Zest Labs developed an IoT-based solution that uses sensors to monitor the freshness of produce from harvest to retail shelf. The system collects data on temperature, humidity, and other factors, using BD analytics to predict shelf life and reduce food waste (https://www.zestlabs.com/zest-fresh-for-produce/, accessed on 27 September 2024). The application of IoT in food quality monitoring has been extensively studied in recent years. Tsang et al. [124] proposed an intelligent model for assuring food quality in managing a multi-temperature food distribution center, demonstrating the practical application of IoT and sensor networks in food supply chain management.

However, the expensive cost of advanced sensors and the massive data generated by IoT systems require complex data management and analysis capabilities [116], which may exceed the capabilities of small food producers. These applications and challenges pose new requirements for food science education. Curriculum reform requires the introduction of IoT and sensor technology in the context of food quality monitoring. Students need to learn how to design sensor networks, manage data streams, and develop algorithms for real-time quality assessment [125]. The course should also cover data security and privacy protection, enabling students to design IoT systems that are both efficient and secure. Considering the trend of integration between IoT technology and other technologies, the course should include case studies on the integration of IoT, blockchain, and AI technologies, such as feeding IoT sensor data into a blockchain system in real-time and using AI algorithms for predictive analysis [126]. In addition, the course should cultivate students’ system integration skills, enabling them to seamlessly integrate IoT solutions with existing food industry systems.

### 4.5. AI-Driven Food Fraud Detection

AI and machine learning algorithms can analyze large datasets to detect patterns indicative of food fraud. Advanced spectroscopic techniques combined with machine learning, similar to those used by Bia Analytical, can rapidly analyze food samples and compare them against a database of authentic products to identify adulterations or mislabeling [127]. Recent research has further advanced the field of AI-driven food fraud detection. Hassoun et al. [128] reviewed the advances in emerging spectroscopic detection methods for food fraud over the past five years, highlighting the role of AI in improving detection accuracy.

However, AI driven food fraud detection faces challenges, such as the need for a large amount of high-quality datasets to train AI models, the interpretability of AI decisions (especially in regulatory environments), and the adversarial attacks that AI systems may face [129]. In addition, with the continuous evolution of food fraud technology, AI systems need to be continuously updated to detect new forms of adulteration. Curriculum reform requires the introduction of AI and machine learning applications in the context of food identification and authenticity verification. Students should learn how to develop and train models for detecting food fraud, including processing various types of data, such as spectral and chromatographic data [130]. The curriculum should also cover AI ethics and decision interpretability, enabling students to design AI systems that are both effective and responsible. Considering the rapid development of food fraud detection technology, the curriculum should cultivate students’ ability to continue learning, so that they can keep up with the pace of technological development. In addition, the course should include case studies on the integration of AI with other technologies, such as blockchain and IoT, exploring how to build a comprehensive food authenticity assurance system.

### 4.6. Other Applications of BD and Blockchain Technology in the Food Industry and the Combined Impact on Curriculum Change

In addition to the many contributions in food testing mentioned above, the convergence of BD and blockchain technologies with the food industry is transforming food safety management, market development, supply chain management, and product innovation. These technological advances are driving the need for significant educational reforms, particularly in food science and related fields. Table 4 lists relevant case studies and impacts on educational curricula.

Future educational curricula must focus on the following areas:Interdisciplinary learning: Courses should combine subjects, such as BD analytics, blockchain technology, food science, agricultural management, and marketing, allowing students to acquire a broad skill set applicable across the food industry [131].Case-driven practical learning: Universities should collaborate with food companies and technology firms to incorporate real-world case studies into their curriculum, allowing students to learn how to apply BD and blockchain technology to solve real-world problems [132].Technological proficiency: Students must be proficient in the use of BD tools and blockchain platforms, enabling them to handle complex datasets and ensure food safety in modern supply chains [133].

By implementing these reforms, educational institutions can ensure that graduates are well-equipped to navigate the technologically advanced landscape of modern food testing and the broader food industry, contributing to enhanced food safety, quality control, and innovation practices worldwide.

## 5. Conclusions

BD and blockchain technology are revolutionizing food safety, supply chain management, and product innovation in the food industry, offering enhanced transparency, efficiency, and quality control. These advancements necessitate a comprehensive overhaul of food science and food safety curricula. Integrating big data analytics and blockchain technology into educational programs will equip students with the skills to navigate and contribute to the future of the food industry. This study, while providing valuable insights, has inherent limitations. The reliance on published scientific literature and case studies may not fully capture all ongoing developments in this rapidly evolving field. Implementation of the proposed educational reforms may encounter challenges, such as resource constraints, faculty expertise gaps, and varying regional regulations. Furthermore, the long-term efficacy of suggested curriculum changes would require longitudinal studies for validation. Future research should address these limitations through empirical studies on big data and blockchain implementation in food testing, coupled with evaluations of educational reform outcomes in this domain.

Notwithstanding these constraints, this study offers a comprehensive overview of the current state and future directions of big data and blockchain applications in food testing. It provides valuable guidance for industry practitioners and educational institutions adapting to the technological revolution in the food industry. Moving forward, educational institutions must continue to adapt their curricula to reflect technological changes in the food industry, ensuring that students are well prepared to meet the complex challenges of modern food safety, transparency, and efficiency. 

## Figures and Tables

**Figure 1 foods-13-03391-f001:**
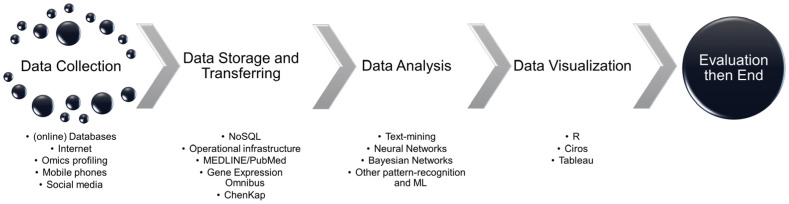
Typical BD case workflow.

**Figure 2 foods-13-03391-f002:**
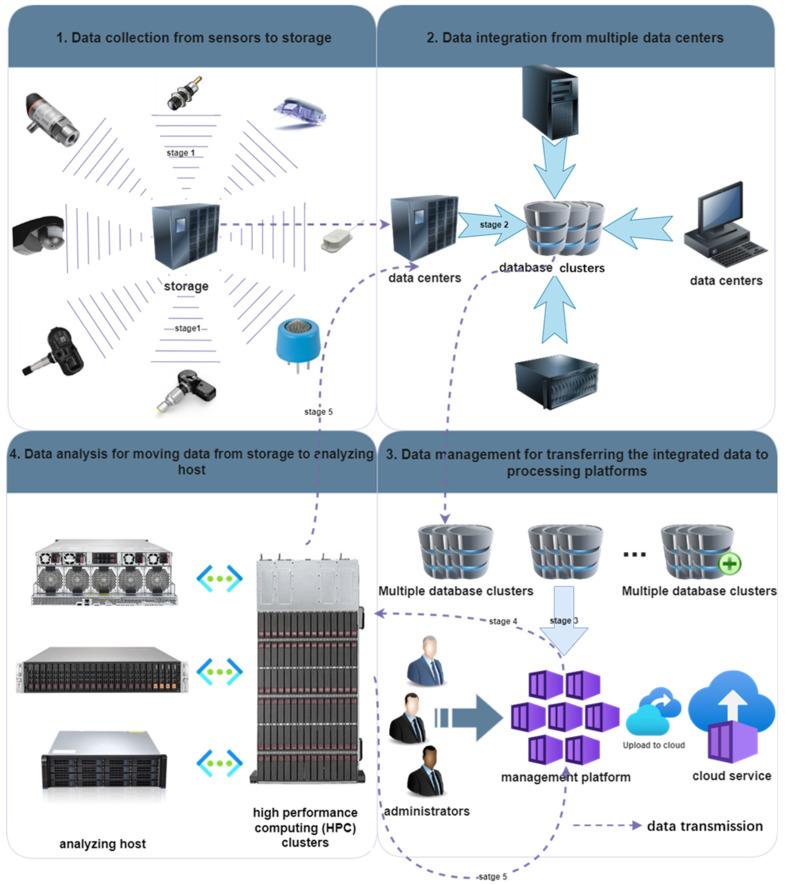
Lifecycle of data transmission [59].

**Figure 3 foods-13-03391-f003:**
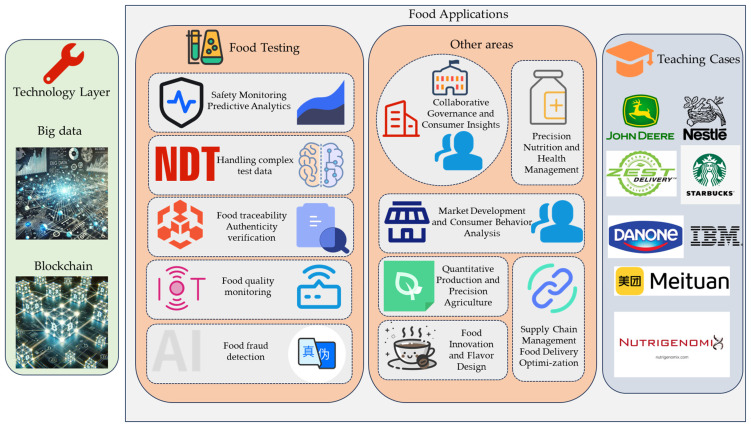
The widespread application and teaching cases of big data and blockchain in the foodindustry.

**Table 1 foods-13-03391-t001:** Online food testing database.

Organization and Country	Database Name	Database Type	Data Description
WHO (Global)	GEMS/food	GEMS/food	Biological/chemical monitoring data
SAMR (China)	SAMR Alerts	Alerts/notifications	Food testing record
USDA-NAL (USA)	USDA National Nutrient Database for Standard Reference	Food product information	Nutrient information food products
European Commission (European Union)	RASFF	Alerts/notifications	Notifications from the Rapid Alert System for Food and Animal Feed
USFDA (USA)	FDA Recent Recalls, Market Withdrawals, and Safety Alerts	Alerts/notifications	FDA Recalls, Market Withdrawals, and Safety Alerts in the last 60 days ^1^
	FDA Archive Recalls, Market Withdrawals, and Safety Alerts		FDA Recalls, Market Withdrawals, and # Safety Alerts ^2^

^1^ Provides information on food recalls, market withdrawals, and safety alerts for the last 60 days. The link (http://www.fda.gov/Safety/Recalls/default.htm, accessed on 21 October 2024) directs to the FDA official recall page, focusing on the latest updates. ^2^ Provides historical recall records for a longer time span. The link (https://www.fda.gov/search?site=FDAgov-recalls&client=FDAgov-recalls&proxystylesheet=FDAgov-recalls&filter=0&getfields=*&q=&requiredfields=recall_category:Food, accessed on 21 October 2024) is to a specific page that searches through the archive of food recall history.

**Table 2 foods-13-03391-t002:** Examples of data analysis methods.

Analysis Method	Analysis Method Type	Database Type Data Description
Image processing algorithm	Convolutional neural network	Appearance defect detection [31]
	Image segmentation (U-Net)	Foreign object detection [32]
Data mining algorithm	Association rule mining (Apriori, FP-Growth)	Consumer behavior analysis [33,34]
	K-means clustering	Security warning [35]
	Decision tree	Classification and grading [36]
Statistical analysis	Regressive analysis	Predicted content and concentration [37]
	Bayesian network	Comprehensive analysis of multiple variables [38]
Natural language processing	Text classification	Food label classification [39]
	Named entity recognition	Extract label information [40]
	Opinion mining	Analyze evaluation and feedback [41]
Machine learning	Summarized by reference [15]	
Recommendation system		
Deep learning	Generative adversarial networks	Food data augmentation [42]
	Autoencoder	Noise removal and data preprocessing [43]
	Long short-term memory	Food quality prediction [44]

**Table 4 foods-13-03391-t004:** Application cases and educational implication of BD and Blockchain in the Food Industry.

Food Industry Application	Case Study	Educational Implication
Collaborative Governance and Consumer Insights	Danone’s flavor development using BD analytics (https://www.danone.com/brands/dairy-plant-based-products/research-and-innovation.html, accessed on 28 September 2024)	Incorporate crowdsourcing and BD analysis tools in food science curricula, teaching students how to collect and analyze consumer feedback for product development
Market Development and Consumer Behavior Analysis	Starbucks’ social media analysis for product adjustments (https://d3.harvard.edu/platform-digit/submission/starbucks-leveraging-big-data-and-artificial-intelligence-to-improve-experience-and-performance/, accessed on 15 September 2024)	Introduce BD tools for clustering analysis and consumer behavior analytics in marketing strategy courses
Quantitative Production and Precision Agriculture	John Deere’s precision agriculture solutions (https://www.deere.com/en/technology-products/precision-ag-technology/, accessed on 28 September 2024)	Include modules on BD in agricultural decision-making, teaching students to analyze real-world agricultural data for production optimization
Food Innovation and Flavor Design	McCormick and IBM’s AI-driven flavor development (https://ir.mccormick.com/news-releases/news-release-details/mccormick-company-and-ibm-announce-collaboration-pioneering-use, accessed on 29 September 2024)	Combine flavor design with BD analysis in food science programs, allowing students to use AI tools for new product development
Supply Chain Management and Food Delivery Optimization	Meituan’s BD-driven delivery time prediction (https://about.meituan.com/en/report/csr-report-2019.pdf, accessed on 29 September 2024)	Integrate BD analytics into supply chain management courses, focusing on efficiency improvement and risk management
Precision Nutrition and Health Management	Nutrigenomix’s personalized nutrition (https://www.nutrigenomix.com/, accessed on 30 September 2024)	Add the application module of personalized nutrition and BD in dietary planning to the course, learn how to interpret genetic and metabolic data to develop personalized nutrition strategies

## Data Availability

No new data were created or analyzed in this study. Data sharing is not applicable to this article.

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
