# Peer review of "Applications of Big Data and Blockchain Technology in Food Testing and Their Exploration on Educational Reform"

_foods, 2024, doi:10.3390/foods13213391_

Round 1

Reviewer 1 Report

Comments and Suggestions for Authors

Comment 1: Lines 29-33: To provide greater clarity within the text, it is proposed that the definition of food safety be presented initially, followed by the food testing mentioned to guarantee it and the generation of big datasets.

Comment 2: Lines 33-35: The term "food safety" is referenced on three occasions within a continuous text passage. Would it be possible to utilize a synonym or link the sentences together?

Comment 3: Lines 41-59: References in the text are missing to support your statements.

Comment 4: Line 78: In the first mention of AI, it is preferable to provide a full definition and to abbreviate the acronym to parenthesis.

Comment 5: Lines 141-147: A reference to Table 2 is missing in the text. Table 2 provides an overview of other approaches that have been proposed for the analysis of big data.

Comment 6: Line 196: ML’s definition is not mentioned in the text.

Comment 7: Lines 287-288: The text mentioned that blockchain technology has been used in various fields. However, the paragraph concludes with a single reference to another review article.

Comments on the Quality of English Language

Point 1: English quality is good.

Reviewer 2 Report

Comments and Suggestions for Authors

1.       blockchains still face challenges in food testing”. It is possible to think that blockchains are directly involved in food testing. How do blockchains provide food testing?

2.       “Furthermore, this study explores the impact of BD and blockchain technologies on educational reform, particularly in food science”. Are food testing and educational reform related?

3.       No reference to table 2.

4.       “Hyperledger Fabric allows any user to join a permissionless blockchain without authorization, and information … [66]”. Is it true? “Hyperledger Fabric, created and supported by the Linux Foundation and IBM, is one of the most popular open-source blockchain permissioned platforms”!! Citation provided is from the same resource [66].

5.       “Future work will emphasize the design principles of channels, which is a key element in enhancing system accessibility [67].”. I cannot make such conclusion based on [67]. Moreover, provide firstly more information on channels.

6.       Provide author names next to reference numbers in Table 3.

7.       materials).Recent” – typo.

8.       Recent research has” – a reference number is missing for the recent research.

9.       “Curricula should incorporate courses on data analytics, IoT applications, and predictive modeling specific to food safety.”. It is obvious. It is done without your consideration.

10.   No meaning to provide directives for educational reform. The analysis of the curricula must be done firstly. Moreover, the curricula have small differences in different countries. Only then it is possible to provide suggestions. All the material concerning educational reform must be removed. Everyone in educational industry is eager to provide advanced material for students. Tight competition exist among universities.

Comments on the Quality of English Language

Moderate editing is required to improve quality of English.

Reviewer 3 Report

Comments and Suggestions for Authors

Thank you for inviting me to review this manuscript, titled “Applications of Big Data and Blockchain Technology in Food Testing and Their Exploration of Educational Reform”. The paper reviews the applications of Big Data (BD) and blockchain technology in modern food testing and explores their impact on educational reform. The research topic proposed by the authors is current and relevant.

Please find my detailed comments below:
In the introduction I would recommend an explicit answer to the following questions: what are the gaps in research? why is this paper interesting?.
The research design, questions, hypotheses and methods are not stated.
The authors investigate the application of each phase in food testing, highlighting the specific challenges: low data collection efficiency and poor data quality; data silos, high data storage costs and inefficient data transfer; high data complexity and challenge of diversity in data analysis methods; difficulty in real-time visualization of unstructured data. The importance of the application of blockchain technology in food testing is revealed, addressing aspects such as: real-time food safety monitoring and predictive analytics, rapid and non-destructive food testing techniques, blockchain for food traceability and authenticity verification, IoT and sensor networks in food quality monitoring, AI driven food fraud detection and other applications of BD and blockchain technology in the food industry. Furthermore, for each issue the relevant educational implications are mentioned.  
No study limitations are presented.  
The work is of real interest, I would suggest an approach corresponding to the journal's requirements.

Best regards

Round 2

Reviewer 2 Report

Comments and Suggestions for Authors

Thank you for the revision.

Several typos can be observed. Some of them are as follows:

                     “environment. [83]. Information”

                     “modes baHHsed on”

Reviewer 3 Report

Comments and Suggestions for Authors

The authors have addressed all of the queries and provided the modifications of the manuscript

Best regards